# Seedling Selection in Olive Breeding Progenies

**DOI:** 10.3390/plants11091195

**Published:** 2022-04-28

**Authors:** Hande Yılmaz-Düzyaman, Raúl de la Rosa, Lorenzo León

**Affiliations:** IFAPA Centro “Alameda del Obispo”, 14004 Córdoba, Spain; raul.rosa@juntadeandalucia.es

**Keywords:** fruit weight, juvenile period, oil content, *Olea europaea*

## Abstract

The long juvenile period in olive (*Olea europaea* L.) delays the evaluation of characters of interest and prolongs the selection of new cultivars in the breeding programs. Therefore, it is important to use accurate selection criteria and appropriate selection pressure to make an effective identification of the superior genotypes and to identify which parents transmit lower juvenile periods to their descendants. In this study, the juvenile period, vigor, fruit fresh weight, and oil content of fruit on a dry weight basis were evaluated in 1568 genotypes from two independent open-pollinated populations; G07, that included 520 genotypes from 25 cultivars and 1 breeding selection and G14, with 1048 genotypes from 13 cultivars. This evaluation was used to test different selection criteria and define optimal selection pressure at the initial stage of an olive breeding program. Wide ranges of variation were obtained for all the characters measured, with higher variability within progenies than between progenies. “Askal” and “Barnea” seem to be the cultivars transmitting the shorter juvenile period to the descendants. In the case of fruit fresh weight and oil content, transgression of variability limits of the parents was observed. Significant correlation was found between mean values of fruit fresh weight of progenies and their parents for G07 (0.59) and G14 (0.95). Selection was made using two selection index formulas (*SI1* and *SI2*). A high coincidence was found between the individuals selected by both formulas and the correspondent selection pressures applied; 15% for *SI1*, and 14% for *SI2*. A wide variability in the percentage of selected genotypes was found, from no individuals selected from some progenies to more than 20% of genotypes selected in some others. These results underline the need to explore the wide genetic variability currently hosted in germplasm collections for an optimal choice of parents in olive breeding works.

## 1. Introduction

Many different olive cultivars arising from domestication over centuries are currently cultivated around the world. This genetic patrimony is preserved in three international germplasm banks in Córdoba (Spain), Marrakech (Morocco), and Izmir (Turkey). The oldest and largest cultivar-hosting germplasm bank is located in Cordoba IFAPA “Alameda del Obispo” and hosts approximately 700 cultivars from 29 countries [1].

However, changes in growing techniques in recent years have underlined the need to obtain new cultivars adapted to modern olive growing. As an example, most traditional cultivars are hardly adapted to modern high-density hedgerow growing, and only a few of them, such as “Arbequina” and “Arbosana”, have been widely planted under this system [2,3]. This situation has promoted the development of breeding programs in different countries, but few new cultivars have been released so far [4,5,6,7,8,9,10].

The juvenile period (JP), i.e., the period of time when seedlings cannot be induced to flower, was traditionally considered the main drawback for breeding olive cultivars, as with other fruit tree species [11]. This long period prolongs the duration of the breeding work and delays the development of new olive cultivars. Different methods were proposed for shortening the juvenile period of the olive tree. Thus, a complete forcing growth protocol of seedlings was developed in our breeding program by [12] based on previous developments in olive and other fruit tree species [13,14]. All these methods encourage the plant to quicky reach the minimum size needed to attain the adult phase [15].

Based on these works, new studies allowed for a better characterization of the relationship between the length of the juvenile period and initial vigor of the seedlings, and the development of early selection criteria for shortening the juvenile period based on vigor trait characterization of seedlings. Among them, trunk diameter and plant height are the most common vigor parameters used for selection [15,16,17], some other more complex tree architectural traits, especially “branch orientation”, together with plant height, have also been proposed as a more effective method in the early selection of seedlings with shorter juvenile periods [18].

A significant influence of the genitors on the juvenile period of their descendants has been observed [19,20]. Therefore, an adequate choice of genitors has also been proposed as a method to shorten this juvenile phase.

Overcoming the juvenile period allows for the evaluation of fruit traits, some of which are considered common main objectives for olive breeding. Thus, early bearing, high productivity, fruit size, and oil content are considered some of the main criteria in current breeding works, usually complemented with several other traits related to disease resistance, oil quality, and suitability for different cultivation systems and mechanical harvesting [10,14,21,22,23,24]. The above-mentioned fruit traits represent, therefore, early selection criteria commonly used in practically all olive breeding efforts. However, the extent to which selection pressure should be applied has been scarcely reported.

Selection at the seedling step directly followed by the final step of multi-environmental trials was the initial strategy followed in our breeding program carried out in Córdoba. However, with this method, a very high selection pressure was applied, and the number of genotypes was reduced drastically to approximately 2% [25]. Afterwards, a potential interest in implementing an intermediate step between the initial seedling evaluation and the final multi-environmental trial was suggested [24]. This was considered a better strategy to reduce selection pressure in the initial step and the risk of discarding potentially interesting genotypes. In this way, selection of around 5–10% of the initial population was suggested as a tentative approach at the initial selection step in full seedling progenies. However, although several olive breeding works have been reported in the last years [9,10,26] information regarding the breeding strategy followed and the selection pressure applied is still quite limited.

In this work, the evaluation of the characters of interest (juvenile period, vigor, fruit fresh weight and oil content) was carried out in two independent open-pollinated populations in order to test different selection criteria and define optimal selection pressure at the initial stage of olive breeding programs.

## 2. Results

### 2.1. Vigor and Juvenile Period

A quite similar distribution of genotypes according to the length of the juvenile period was observed in both generations evaluated (Figure 1). Around 7% of the total population, in both G07 and G14, showed short juvenile period starting flowering and bearing fruits three years after planting. On the opposite, 29% and 18% of the genotypes from G07 and G14 generations, respectively, showed remarkable long juvenile period with no flowering observed six years after planting.

A wide range of variation was also found for trunk diameter measurements one meter height from the ground taken four years after planting (29–99 mm). Significant differences were found between progenies in terms of both vigor and length of the juvenile period (Table 1 and Table 2).

A general negative association between vigor and length of the juvenile period was observed, i.e., the genotypes with low vigor showed longer juvenile period. Thus, significant differences in plant vigor were obtained according to the length of the juvenile period in both generations (Figure 2). In fact, the progenies of the cultivars “Askal” and “Barnea”, evaluated in G14, showed the shortest juvenile period and were among the most vigorous (Table 2). Similarly, the progenies of “Toffahi”, “Canetera” and “Hojiblanca” were also interesting in terms of their relation between shorter juvenile period and high vigor among the other progenies evaluated in G07 (Table 1). One exception to this relation was “Frantoio” that showed high vigor and high juvenile period. On the contrary, progenies from “Chorruo Castro Río” and “Meski” in G07, and “Cordobes Arroyo Luz” in G14 were characterized by low vigor and longer juvenile period.

Finally, it should be noted that no significant correlation was obtained between mean values of progenies and their parents for both plant vigor four years after planting (data not shown).

### 2.2. Variability for Fruit Fresh Weight and Oil Content of Fruit in Dry Weight

A wide range of variation was obtained for fruit fresh weight (FFW) and oil content of fruit in dry weight (OCFrDW) among the genotypes evaluated. The parents showed higher mean values for FFW and OCFrDW in both generations (Table 3). A similar coefficient and range of variation was obtained from progenies than the parents for FFW, while a clearly wider range of variation in progenies than the corresponding parents was observed for OCFrDW in both generations.

The progenies of “Chalkidiki” (G07) and “Mahati-1010” (G14) had the highest means in terms of FFW. Besides, the progenies of “Leccino”, “Negrillo de Arjona”, “Manzanilla de Sevilla” from G07 and “Askal” from G14 had the highest means for OCFrDW. The lowest means for FFW and OCFrDW were obtained from “Lechín de Granada” (G07) and “Cordovil de Serpa-130” (G14). The progenies of “Meski” cultivar were not evaluated for the traits FFW and OCFrDW because the few seedlings reaching the adult phase did not produce enough number of fruits (Table 1 and Table 2).

The means of the progenies of “Arbequina”, “Blanqueta”, “Canetera”, “Changlot Real”, “Empeltre”, “Frantoio”, “Koroneiki” and “Sikitita” from G07, “Askal”, “Barnea” and “Grosal Vimbodi” from G14 were higher than their parents in terms of FFW. Likewise, for OCFrDW, progenies of “Blanqueta”, “Canetera”, “Hojiblanca”, “Leccino”, “Manzanilla de Piquito”, “Morona”, “Toffahi” from G07, and “Askal”, “Cordobés Arroyo Luz”, “Dulzal” and “Mahati-1010” from G14 showed higher means.

A transgression of variability limits of the female parent was obtained in most cases. Except the progenies of the cultivars “Leccino”, “Morona”, “Ocal”, “Picudo”, “Tanche”, “Toffahi” from G07, numerous seedlings showed higher values than their female parent for FFW. In G14, seedlings superior to their parents were found from the offspring of the different cultivars evaluated, except for “Acebuchera”, which could not be evaluated due to lack of parental data. Similarly, for OCFrDW, individuals with higher values were obtained from all evaluated progeny of different parents except “9-67”, “Chalkidiki”, “Trylia” in G07 and “Morisca” in G14 (Table 1 and Table 2).

Significant differences among the progenies were obtained for the fruit characteristics analyzed. The variance within progenies was found to be greater than between progenies in both generations, with higher contribution for OCFrDW than for FFW (Table 4).

Significant correlation was obtained between mean values of FFW of progenies and their parents for G07 and G14, while lower (G07) or non-significant (G14) correlation values were observed for OCFrDW (Figure 3).

### 2.3. Selection Practise

Selection was performed using two selection index formulas. Most of the selected genotypes using both formulations were the same. They constituted 15% of the total population using the *SI1* formula and 14% using the *SI2* formula (Table 5). When comparing G07 and G14, the percentage of individuals selected from both generations were the same with the *SI1* formula (15%) and very similar with the *SI2* formula, 13% and 14%, respectively.

Wide variability between progenies was obtained according to the percentage of genotypes finally selected (Table 5). No individuals were selected from the progenies of the “Chorruo de Castro del Río” and “Lechín de Granada” progenitors by either formula. On the contrary, more than 20% of genotypes were selected with both formulas from the progenies of “Arbequina”, “Canetera”, “Chalkidiki”, “Hojiblanca”, “Manzanilla de Sevilla” and “Toffahi” in G07, and “Askal” and “Cornezuelo de Jaen” in G14. Most of the selected genotypes showed intermediate vigor values measured as trunk diameter one meter height from the ground taken four years after planting, 76% between 50–75 mm, with 8 and 16% of them showing lower or higher values respectively.

## 3. Discussion

In the present study, 29% and 18% of the genotypes from G07 and G14, respectively, did not flower even after 6 years, while only 7% of the seedlings showed short juvenile periods and, therefore, started flowering and bearing fruits three years after planting. However, shorter juvenile periods for a higher number of genotypes have been reported in previous studies. First flowering occurred for 29% of the individuals approximately 2.5 years after germination, while only 7% of the genotypes had long juvenile periods with no flowering even 6 years after germination [12]. In another population of 278 genotypes derived from three combinations and one open-pollination, 1% of the seedlings had juvenile period of 2 years, 15% with 3 years, 34.6% with 4 years, and 27.9% with 5 years and the remainder of the population had a juvenile period of 6 years or more [17]. Similarly, in a population of 794 genotypes obtained from five crosses and one open-pollination, 3% of flowering occurred at the 2nd year, 18% at the 3rd and 15% at the 4th year [18]. The longer general juvenile period observed in this work compared to previous reports in olive could be associated to a general slower development of plants. In this study, the average vigor measured as trunk diameter one meter height from the ground, taken four years after planting (55 and 58 mm for G07 and G14, respectively), was much lower than previously reported at similar seedling age [12]. Forcing growth protocols, cultural practices, and both field and environmental conditions are difficult to standardize and, therefore, remarkable differences in plant growth could be expected.

Significant negative correlations were previously reported between shorter juvenile periods and different plant vigor and architectural parameters, such as stem diameter, plant height, and secondary shoots insertion angle in olive as well as in other fruit tree species [16,17,18,27,28]. A similar negative correlation between juvenile period and vigor was obtained in this study. The fact that “Frantoio” did not follow this correlation can be explained by the significant high vigor of this cultivar [25]. Besides, cumulative crop of three years (3, 4, and 5 years after planting) was observed to be higher in genotypes with a short juvenile period, which is in accordance with previous findings [16].

The results of the evaluation in two independent open-pollinated populations carried out in this work showed similar and wide ranges of variation than previously reported in other populations both for FFW and OCFrDW. Thus, in previous studies higher mean values were obtained from cultivars as compared to their progenies [29], and a wider variation in progenies as compared to the cultivars [23,29]. Similarly, the results of this study showed higher mean values for the parents than that of the progenies. Examining the open-pollinated progenies of 17 Spanish cultivars, [23] found almost the same variations for both FFW ranging from 0.44 to 11.78 g and OCFrDW ranging from 7.80% to 54.99%. However, our study showed higher maximum values for OCFrDW in both progenies than the one observed in that previous study. A wide variability for both FFW and OCFrDW has been obtained even in progenies from a single cross combination between “Oliviere” × “Arbequina” [30]. The variance we found for both FFW and OCFrDW within progenies was higher than between progenies was also in accordance with previous works [29]. Consistent with other studies, many offspring were found to have higher values for FFW and OCFrDW compared to their parents, i.e., a high number of transgressive segregants was found for these traits [23,29,30,31]. Therefore, it can be stated that open-pollination is an effective strategy to obtain new breeding genotypes with better agronomic characteristics than their parents [23,30], as also suggested in other fruit tree breeding programs [32].

Highly significant correlation between mean values of progenies and their parents was obtained for FFW, while for OCFrDW correlation values were lower or non-signifcant. In previous works in olive, both significantly positive [23] and negative correlations [31] between parental means and their progenies have been reported for FFW and OCFrDW. These results suggest that a different inheritance could be expected for different traits and populations, and further research will be necessary to elucidate the heritability for these traits in olive.

In fruit breeding programs, it has been underlined the importance of reducing as much as possible the number of genotypes under evaluation at the initial populations due to reasons such as decreasing labor and cost and making a more detailed evaluation of the remaining selected genotypes. In an olive breeding study conducted in Italy, 150 genotypes (3%) were selected from the initial population of 5000 seedlings according to their drupe traits and tree architecture [33]. In another study carried out in Turkey, 23 of 393 (4.7%) genotypes were selected for the next stage of the breeding program considering different purpose such as oil content, fruit weight and oil composition [34]. Dridi et al. [8] selected 13 genotypes of 200 (6.5%) from a breeding program of Tunisia as promising based on their fruit and endocarp weights. Similarly, in the early years of our breeding program, 15 out of 748 seedlings (2%) of the initial population were selected for further multi-environment trials according to early bearing and high early yield [25,29]. These results represent a considerable selection pressure at initial stages and, therefore, a high risk of discarding potentially interesting genotypes. To minimize this risk, a lower selection pressure has been suggested as a possible alternative by introducing an intermediate step of selection between the initial seedling stage and the final multi-environment trials [24].

The results of this work with a selection rate around 13–15% according to different generations and selection indexes, support this previous recommendation to include three evaluation steps in olive breeding programs (initial seedling step, intermediate step, and multi-environmental trials) by applying a lighter selection pressure at the initial seedling stage. In spite of the correlation between juvenile period and trunk diameter, certain vigor variability was still retained among the selected genotypes. This could be interesting for the final selection of different genotypes according to the planned growing system, from high density hedgerow orchards (low vigor genotypes preferred) to traditional low planting density. However, it should be noted that no correlation has been found in previous works for vigor measurements taken at different steps of selection [24] and, therefore, additional evaluation for this trait should be carried in future works. The different percentages of genotypes selected among the different open-pollinated progenies were also remarkable, indicating the need of an adequate choice of parents is of paramount importance for breeding works. In this sense, cultivars “Askal” and “Canetera” combined a high percentage of seedlings with good agronomic traits, with mean progeny values for fruit weight and oil content higher than those of the parent cultivar. This makes those cultivars very interesting as genitors in breeding programs. This information should be confirmed by the evaluation of further agronomic traits as growth habit, biennial bearing and productivity in further evaluation steps.

## 4. Materials and Methods

### 4.1. Plant Materials

A total of 1568 genotypes obtained in two different years, namely 2007 (G07) and 2014 (G14), were evaluated in this study. These genotypes derived from open-pollination of 38 cultivars from the Córdoba World Olive Germplasm Bank (WOGBC) at IFAPA [35] (Belaj et al., 2018) plus one genotype previously selected in the breeding program of IFAPA Córdoba. G07 consists of 520 genotypes obtained from open-pollination of 25 cultivars and one previously selected genotypes: “9–67”, “Arbequina”, “Blanqueta”, “Canetera”, “Chalkidiki”, “Changlot Real”, “Chorruo de Castro del Río”, “Empeltre”, “Frantoio”, “Hojiblanca”, “Koroneiki”, “Leccino”, “Lechín de Granada”, “Lechín de Sevilla”, “Manzanilla de Sevilla”, “Manzanilla del Piquito”, “Meski”, “Morona”, “Negrillo de Arjona”, “Ocal”, “Picual”, “Picudo”, “Sikitita”, “Tanche”, “Toffahi”, and “Trylia”. G14 consists of 1048 genotypes obtained from open-pollination of 13 cultivars: “Acebuchera”, “Aloreña de Iznalloz”, “Askal”, “Barnea”, “Bodoquera”, “Cordobes Arroyo Luz”, “Cordovil de Serpa-130”, “Cornezuelo de Jaén”, “Dulzal”, “Grosal Vimbodi”, “Mahati-1010”, “Morisca”, and “Nevadillo de Santisteban del Puerto”. Seedlings were obtained with the standard forced growth protocol applied in olive breeding studies [12] and replicates of the parent cultivars were also obtained by vegetative propagation by semi hardwood stem cuttings. All plants were transplanted to the experimental field of IFAPA Córdoba at 4 m × 1.5 m spacing and subjected to standard cultural practices.

### 4.2. Traits Evaluated

At the beginning of the harvest season in the 3rd, 4th and 5th years after planting (YAP), crop of each tree was qualitatively evaluated with assigned values from 0 (no crop) to 3 (high crop) for both generations. The total crop was calculated by summing these three years’ values for each tree. Fruit samples from each tree were randomly collected and three sets of subsamples of approximately 25 g each and the same number of fruits were selected for measuring fruit fresh weight (FFW). The samples were then dried in a forced-oven at 105 °C for 42 h until completely dehydrated. The oil content was determined on a fruit dry weight basis (OCFrDW) with the NMR fat analyzer. FFW and OCFrDW values were recorded 4YAP and 5YAP, and the mean of these two years were used for evaluation of each character. The trunk diameter one meter height from the ground was measured four years after plating in the field as an expression of plant vigor.

### 4.3. Selection Practice

Total crop, fruit fresh weight and oil content in dry weight basis were used for selection of the genotypes. For this purpose, normalized data of total crop, FFW and OCFrDW were used to be able to compare the two generations, G07 and G14. The normalized total crop for each individual was obtained by dividing the difference between the total crop of each individual and the mean of total crop of the progenies by the standard deviation of the progenies. Normalized FFW and OCFrDW values of each individual were calculated separately for generations G07 and G14 by using the mean of the corresponding parents and the standard deviation of the same parents. To understand if the different contribution percentage of each character affects the final selected individuals, two different selection index formulas (*SI1* and *SI2*) were applied by using the normalized data. Each formula tries to combine the three characters with a different percentage of participation of each one of them. *SI1* and *SI2* values of each genotype were calculated as follows:*SI1* = 33.3% FFW + 33.3% OCFrDW + 33.3% Total crop,
*SI2* = 25% FFW + 50% OCFrDW + 25% Total crop

*SI1* arose from the assumption that the percentage contribution of each character is equally important. *SI2*, on the other hand, was prepared with the assumption that the content of oil in dry basis is 50% important in the selection of genotypes, since one of the fundamental objectives of the breeding program is to develop new cultivars with high oil yield. In any case, all individuals with a total crop lower than the mean of the whole progenies were discarded. The similarity of selected individuals from each generation and the selection index was compared.

### 4.4. Data Analysis

Variability of fruit traits and vigor were evaluated in two independent open-pollinated initial populations (G07 and G14). Analysis of variance was performed for FFW, OCFrDW, juvenile period and vigor in both G07 and G14. Sources of variation were calculated by partitioning the sum of squares. Correlations among all the traits and between parents and progenies were calculated. Normalized data were obtained for selection practice and selection was performed with the two different formulas above mentioned.

## 5. Conclusions

In summary, the results of the evaluation in two independent open-pollinated populations carried out in this work are consistent with previous findings in olive in terms of variability and relationships among interesting traits. The results obtained from different generations and selection indexes suggest and optimal selection rate around 15% at this initial stage of the breeding process, and support previous recommendation to include three evaluation steps in olive breeding programs. Finally, the highly different percentage of potentially interesting genotypes selected among the different open-pollinated progenies underlines the need to explore the wide genetic variability currently hosted in olive germplasm collections. An optimal choice of parents in olive breeding works would facilitate obtaining new olive cultivars specifically adapted to the current trends of olive growing.

## Figures and Tables

**Figure 1 plants-11-01195-f001:**
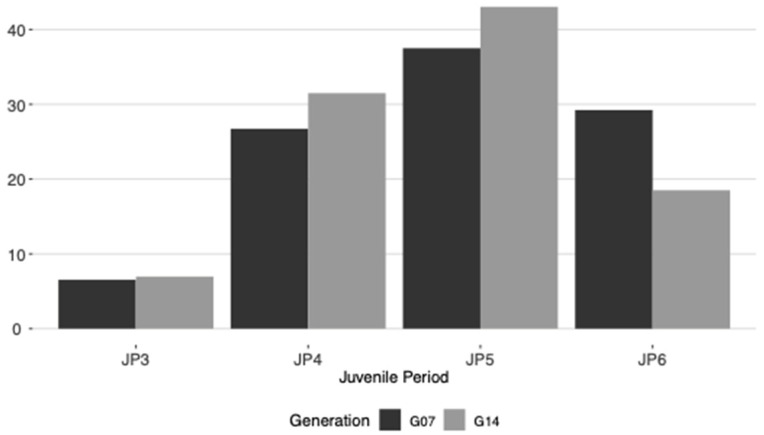
Distribution of genotypes according to the length of the juvenile period (JP, years until reaching adult phase and flowering) in G07 and G14.

**Figure 2 plants-11-01195-f002:**
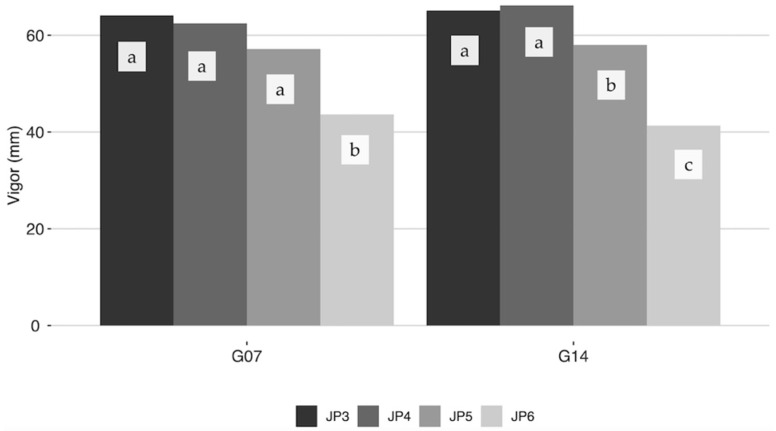
Vigor (mm) of the seedlings four year after planting according to the length of the juvenile period in G07 and G14. Different letters (a to c) indicate significant differences (*p* < 0.001) between vigor of the seedlings overcoming juvenility 3 (JP3), 4 (JP4), 5 (JP5) and 6 (JP6) years after planting.

**Figure 3 plants-11-01195-f003:**
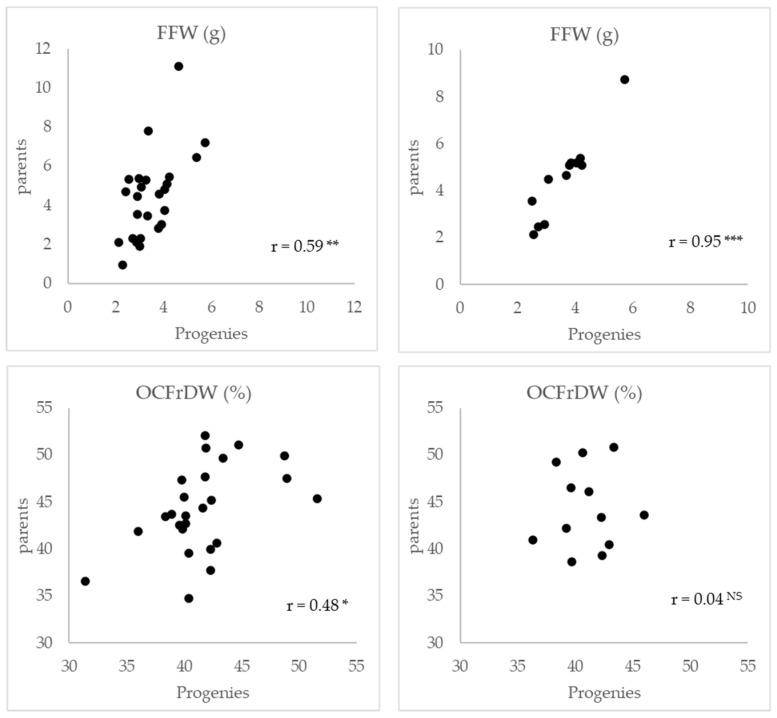
Correlations between averages values for FFW and OCFrDW in progenies and their corresponding parents G07 and G14. ^NS^, *, **, ***: non-significant and significant at *p* < 0.05, 0.01, 0.001 respectively.

**Table 1 plants-11-01195-t001:** Main agronomic traits of the open-pollinated G07 progeny.

	JP	TD	FFW					OCFrDW		
	Progeny	Progeny	Parent	Progeny			Parent	Progeny		
	Mean	Mean		Mean	Min	Max		Mean	Min	Max
“9–67”	5.0	48.7	3.5	2.9	1.8	4.3	50.8	41.9	32.0	49.6
“Arbequina”	4.6	58.0	1.9	3.0	1.8	4.5	43.7	38.9	16.8	47.7
“Blanqueta”	4.9	55.1	2.1	2.9	1.0	5.9	37.8	42.3	35.0	52.5
“Canetera”	4.5	63.5	3.0	3.9	1.6	10.2	40.0	42.3	39.0	46.4
“Chalkidiki”	4.9	57.6	7.2	5.8	2.5	9.8	51.1	44.7	35.7	49.6
“Changlot Real”	5.1	50.7	3.8	4.0	2.4	5.9	47.7	41.8	31.4	53.6
“Chorruo Castro Río”	5.4	40.4	6.5	5.4	2.2	8.8	41.9	36.0	16.8	54.3
“Empeltre”	4.9	64.6	2.8	3.8	2.7	5.8	45.6	40.0	30.7	47.7
“Frantoio”	5.2	64.9	2.3	3.0	2.2	4.6	45.2	42.4	36.0	45.0
“Hojiblanca”	4.5	60.2	4.8	4.1	2.2	8.0	40.7	42.8	34.8	50.6
“Koroneiki”	4.6	59.0	1.0	2.3	1.2	3.9	43.6	40.2	28.1	48.2
“Leccino”	5.1	48.9	5.3	2.6	2.6	2.6	45.4	51.6	51.6	51.6
“Lechín de Granada”	5.2	47.9	2.1	2.1	1.1	3.4	36.6	31.4	7.8	43.0
“Lechín de Sevilla”	5.3	56.2	3.4	3.3	2.3	5.1	42.2	39.9	22.9	49.4
“Manzanilla de Sevilla”	4.5	53.3	5.5	4.3	2.5	6.2	49.9	48.7	42.9	51.3
“Manzanilla del Piquito”	4.8	55.7	5.3	3.3	2.0	5.1	39.6	40.4	23.3	46.6
“Meski”	5.5	47.9	7.5				47.3			
“Morona”	4.8	56.8	5.4	3.0	2.1	3.9	42.7	40.2	30.6	49.2
“Negrillo de Arjona”	5.2	56.5	5.1	4.2	3.2	5.1	47.5	48.9	41.7	56.1
“Ocal”	4.9	49.2	7.8	3.4	1.5	7.1	52.1	41.9	27.4	53.4
“Picual”	4.7	59.4	4.5	2.9	0.5	5.0	47.4	39.8	14.1	49.7
“Picudo”	4.7	63.6	4.9	3.1	1.8	4.0	42.6	39.6	31.8	47.5
“Sikitita”	5.1	38.2	2.3	2.7	1.8	4.2	44.4	41.6	36.9	47.7
“Tanche”	4.8	60.7	4.7	2.4	1.6	4.3	43.4	38.4	27.9	46.7
“Toffahi”	4.5	58.7	11.1	4.6	3.1	5.9	34.8	40.4	28.5	48.1
“Trylia”	5.0	53.7	4.6	3.8	2.5	5.1	49.7	43.4	35.5	48.6

For the progenies, mean values of juvenile period (JP, years), trunk diameter (TD, mm) and mean, maximum (Max) and minimum (Min) values for fruit fresh weight (FFW, g), oil content of fruit in dry weight (OCFrDW, %) are included. For the female parent, FFW and OCFrDW are included. Mean progeny values of FFW and OCFrDW are underlined when higher than the one of the correspondent female parent.

**Table 2 plants-11-01195-t002:** Main agronomic traits of the open-pollinated G14 progeny.

	JP	TD	FFW					OCFrDW		
	Progeny	Progeny	Parent	Progeny			Parent	Progeny		
	Mean	Mean		Mean	Min	Max		Mean	Min	Max
“Acebuchera”	4.5	57.3		2.9	1.5	4.1		43.5	27.1	49.8
“Aloreña de Iznalloz”	4.9	54.3	5.1	3.8	1.7	7.0	50.3	40.6	24.6	51.8
“Askal”	4.2	63.5	2.1	2.6	1.0	5.4	43.7	46.0	28.0	58.5
“Barnea”	4.3	68.1	2.6	2.9	1.4	5.2	46.1	41.2	30.6	55.5
“Bodoquera”	4.7	62.1	5.2	4.0	1.8	7.6	43.4	42.2	25.2	56.6
“Cordobes Arroyo Luz”	5.4	55.5	4.7	3.7	1.3	5.1	40.5	43.0	30.1	50.5
“Cordovil de Serpa-130”	4.6	55.4	3.6	2.5	1.0	5.1	41.0	36.3	22.6	47.2
“Cornezuelo de Jaen”	4.7	52.0	5.4	4.2	1.5	6.4	50.8	43.4	26.3	53.8
“Dulzal”	4.7	54.4	4.5	3.1	1.7	5.8	39.3	42.3	33.1	52.1
“Grosal Vimbodi”	4.9	59.2	2.5	2.7	1.2	6.5	42.3	39.3	28.5	52.5
“Mahati-1010”	4.8	56.5	8.7	5.7	2.2	8.9	38.7	39.7	21.1	46.4
“Morisca”	5.2	57.0	5.2	3.9	1.5	6.4	49.3	38.4	25.8	47.4
“Nevadillo de Sant Pto”	4.9	53.4	5.1	4.2	0.8	8.4	46.6	39.7	25.1	53.1

For the progenies, mean values of juvenile period (JP, years), trunk diameter (TD, mm) and mean, maximum (Max) and minimum (Min) values for fruit fresh weight (FFW, g), oil content of fruit in dry weight (OCFrDW, %) are included. For the female parent, FFW and OCFrDW are included. Mean progeny values of FFW and OCFrDW are underlined when higher than the one of the correspondent female parent.

**Table 3 plants-11-01195-t003:** Mean, coefficient of variation (CV), maximum (Max) and minimum (Min) values for progenies of generations G07 and G14 and their corresponding parents for fruit fresh weight (FFW) and oil content of fruit in dry weight (OCFrDW).

			Mean	CV	Min	Max
FFW (g)	G07	Parents	4.56	49.34	0.97	11.12
		Progenies	3.49	43.55	0.51	10.19
	G14	Parents	4.55	39.12	2.12	8.74
		Progenies	3.35	43.28	0.82	8.93
OCFrDW (%)	G07	Parents	44.37	10.25	34.78	52.14
		Progenies	40.94	17.24	7.8	56.12
	G14	Parents	44.31	9.64	38.66	50.84
		Progenies	41.75	16.86	21.08	58.46

**Table 4 plants-11-01195-t004:** Variance components between (VB) and within (VW) progenies for the traits evaluated in progenies from open-pollination of 38 olive cultivars and one breeding selection.

	Variance Components	FFW (%)	OCFrDW (%)
G07	VB	35.78	20.54
	VW	64.22	79.46
G14	VB	32.81	14.78
	VW	67.19	85.22

**Table 5 plants-11-01195-t005:** Parents of the progenies, number of the genotypes evaluated from each progeny, number of the genotypes selected using *SI1* and *SI2*, and the ratio of selected individuals to the total population (%).

	Progenies	No of Genotypes	*SI1*	%	*SI2*	%
G07	TOTAL	520	77	15	69	13
	“9-67”	21	2	10	2	10
	“Arbequina”	20	4	20	4	20
	“Blanqueta”	21	4	19	4	19
	“Canetera”	20	7	35	5	25
	“Chalkidiki”	20	5	25	5	25
	“Changlot Real”	21	3	14	2	10
	“Chorruo de Castro del Río”	21	-	-	-	-
	“Empeltre”	21	3	14	3	14
	“Frantoio”	21	1	5	1	5
	“Hojiblanca”	20	5	25	4	20
	“Koroneiki”	18	3	17	3	17
	“Leccino”	15	1	7	1	7
	“Lechín de Granada”	18	-	-	-	-
	“Lechín de Sevilla”	21	1	5	1	5
	“Manzanilla de Sevilla”	20	8	40	8	40
	“Manzanilla del Piquito”	21	2	10	2	10
	“Meski”	18	-	-	-	-
	“Morona”	21	3	14	2	10
	“Negrillo de Arjona”	20	1	5	1	5
	“Ocal”	21	3	14	3	14
	“Sikitita”	19	2	11	2	11
	“Picual”	20	4	20	3	15
	“Picudo”	21	3	14	2	10
	“Tanche”	20	2	10	3	15
	“Toffahi”	20	6	30	4	20
	“Trylia”	21	4	19	4	19
G14	TOTAL	1048	156	15	145	14
	“Acebuchera”	76	9	12	9	12
	“Aloreña de Iznalloz”	67	9	13	7	10
	“Askal”	96	35	36	35	36
	“Barnea”	100	20	20	17	17
	“Bodoquera”	88	20	23	17	19
	“Cordobes Arroyo Luz”	81	3	4	3	4
	“Cordovil de Serpa-130”	78	4	5	4	5
	“Cornezuelo de Jaen”	88	20	23	19	22
	“Dulzal”	81	11	14	10	12
	“Grosal Vimbodi”	76	3	4	3	4
	“Mahati-1010”	71	12	17	11	15
	“Morisca”	80	3	4	3	4
	“Nevadillo de Sant Pto”	66	7	11	7	11
	TOTAL	1568	233	15	214	14

## Data Availability

The data presented in this study are available on request from the corresponding author.

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
