# Peer review of "Seedling Selection in Olive Breeding Progenies"

_plants, 2022, doi:10.3390/plants11091195_

Round 1

Reviewer 1 Report

The manuscript entitled: "Seedling selection in olive breeding progenies" aims at studying different selection criteria and defining optimal selection pressure at the initial stage of olive breeding program. the manuscript is clear and well-written. The rationale behind the study supported by the literature and the experimental design well conceived. Nevertheless, reading the manuscript it seems like the final part of the text is missing. Data analysis is very brief and conclusions are missing. In case of error, I suggest to the authors to include the complete manuscript; otherwise, I think they should discuss widely data analysis and providing conclusions

Author Response

The manuscript entitled: "Seedling selection in olive breeding progenies" aims at studying different selection criteria and defining optimal selection pressure at the initial stage of olive breeding program. the manuscript is clear and well-written. The rationale behind the study supported by the literature and the experimental design well conceived. Nevertheless, reading the manuscript it seems like the final part of the text is missing. Data analysis is very brief and conclusions are missing. In case of error, I suggest to the authors to include the complete manuscript; otherwise, I think they should discuss widely data analysis and providing conclusions

Dear reviewer, thank you very much for your comments. Following your recommendations, Results have been commented more in detail and a specific Conclusions section has been included.

Reviewer 2 Report

I find this manuscript quite interesting. I only have a few recommendations that should be considered before publication:

  • Please, provide the full botanical name of the species in the Abstract.
  • The authors are mixing two terms" variety and cultivars - this is not the same, please, correct it.
  • MDPI uses the serial comma.
  • All abbreviations must be explained when first mentioned.
  • Minor grammar and style errors need correction.
  • Some additional references in the Introduction are needed.
  • The duration of the juvenile periods (JP3-JP6) in the Figures should be described better. What does it mean exactly? 
  • Please provide the units in Table captions.
  • I suggest providing a separate Conclusions chapter.

For more specific comments, please see the corrected manuscript.

After incorporating all the necessary changes the manuscript can be accepted for publication.

Author Response

I find this manuscript quite interesting. I only have a few recommendations that should be considered before publication:

Dear reviewer, thank you very much for your comments and corrections for improving the manuscript.

Please, provide the full botanical name of the species in the Abstract. It has been done.

The authors are mixing two terms" variety and cultivars - this is not the same, please, correct it. “Cultivar” has been used throughout the manuscript.

MDPI uses the serial comma. It has been corrected.

All abbreviations must be explained when first mentioned. It has been done.

Minor grammar and style errors need correction. A new revision has been done.

Some additional references in the Introduction are needed. There was a mistaken full stop, new paragraph, that creates confusion at the introduction section. It has been corrected.

The duration of the juvenile periods (JP3-JP6) in the Figures should be described better. What does it mean exactly? Description and additional information regarding JP have been included in all figure captions. Additional information and description have been added both in figures legends and main text.

Please provide the units in Table captions. It has been done.

I suggest providing a separate Conclusions chapter. Following your recommendation and reviewer 1’s, a specific Conclusions section has been included. A final brief conclusion has been also added in the Abstract.

For more specific comments, please see the corrected manuscript. Thanks again for your comments and corrections

Round 2

Reviewer 1 Report

In my opinion, the manuscript is ready for acceptance in its current form